# Agent-Based Modelling of Health Inequalities following the Complexity Turn in Public Health: A Systematic Review

**DOI:** 10.3390/ijerph192416807

**Published:** 2022-12-14

**Authors:** Jennifer Boyd, Rebekah Wilson, Corinna Elsenbroich, Alison Heppenstall, Petra Meier

**Affiliations:** 1MRC/CSO Social and Public Health Sciences Unit, School of Health and Wellbeing, University of Glasgow, Glasgow G3 7HR, UK; 2School of Health and Related Research, University of Sheffield, Sheffield S1 4DA, UK; 3College of Medical, Veterinary and Life Sciences, University of Glasgow, Glasgow G61 1QH, UK; 4School of Social and Political Sciences, University of Glasgow, Glasgow G12 8RT, UK

**Keywords:** health inequality, agent-based modelling, simulation, systematic review

## Abstract

There is an increasing focus on the role of complexity in public health and public policy fields which has brought about a methodological shift towards computational approaches. This includes agent-based modelling (ABM), a method used to simulate individuals, their behaviour and interactions with each other, and their social and physical environment. This paper aims to systematically review the use of ABM to simulate the generation or persistence of health inequalities. PubMed, Scopus, and Web of Science (1 January 2013–15 November 2022) were searched, supplemented with manual reference list searching. Twenty studies were included; fourteen of them described models of health behaviours, most commonly relating to diet (*n* = 7). Six models explored health outcomes, e.g., morbidity, mortality, and depression. All of the included models involved heterogeneous agents and were dynamic, with agents making decisions, growing older, and/or becoming exposed to different health risks. Eighteen models represented physical space and in eleven models, agents interacted with other agents through social networks. ABM is increasingly contributing to our understanding of the socioeconomic inequalities in health. However, to date, the majority of these models focus on the differences in health behaviours. Future research should attempt to investigate the social and economic drivers of health inequalities using ABM.

## 1. Introduction

Systematic socioeconomic inequalities in health persist and continue to widen within many economically prosperous countries across the globe [1,2]. The socioeconomic gradient in health remains one of the main challenges for public health as socioeconomically disadvantaged individuals have a lower life expectancy and a higher risk of developing life-limiting illnesses, such as diabetes and cardiovascular disease, compared to their advantaged counterparts [3,4].

The theories and frameworks developed to understand the causes of and solutions to the socioeconomic gradient in health are undoubtedly complex. For example, the World Health Organization’s (WHO) Commission on the Social Determinants of Health (CSDH) developed a conceptual framework to illustrate the relationship between the social determinants of health and equity in health and wellbeing, which was multi-level and contained feedback loops [5]. The CSDH framework highlights the multi-faceted nature of inequality from the impact of the socioeconomic and political context to psychosocial factors and biology. Thus, there is an increasing recognition that health inequality is a complex or ‘wicked’ problem and systems simulation models are a useful tool to understand the underlying causes and mechanisms [6].

Complex systems are systems which consist of interacting parts or subsystems. Some key characteristics of complex systems are dynamic, resulting in adaptation to change, non-linear relationships, feedback loops, tipping points, and the emergence of macro-phenomena from interactions at the micro level (see, e.g., CECAN 2018) [7]. It is difficult to capture these relationships using a traditional epidemiological “risk factor” approach which uses linear reductionist models to test the relationships between decontextualised dependent and independent variables [8]. Agent-based modelling (ABM), a well-established methodological approach used widely in the field of social science, has been highlighted as a methodological approach that can be used to address this problem [6]. ABM involves simulating the actions and interactions of individual agents with other agents and their environment based on a set of specified rules and observing emergent phenomena [9]. Agents may adapt their own behaviour in response to previous behaviour, their social network, or environmental stimuli [9]. Not only can ABM be used to understand complex phenomena, but they can also be used to test the impact of policy interventions and inform policy decisions and have been successfully applied in other areas of public health, particularly for the control of infectious diseases [10].

ABM has been used successfully to understand the causes of inequality more broadly outside the field of public health. Famously, the Schelling model of segregation which identified residential segregation is generated in the presence of relatively simple nearest neighbour preferences and could be used to understand the racial segregation patterns in the USA [11]. Additionally, the Sugarscape model developed by Epstein and Axtell has offered insights into the generation of wealth inequality using a relatively simple model which simulates the land in which sugar is grown and can be harvested by individuals to become their wealth [12,13]. Individuals in the simulation are programmed to harvest the sugar closest to them; strikingly, even when the wealth available to all individuals at the beginning of the simulation is equal, trends in wealth inequality are produced even after a short simulation period. Additionally, only a very small proportion of individuals have high levels of wealth, while a much larger proportion have low levels of wealth. These models, alongside many others developed in the field of social science, have illustrated the benefits of using ABM to understand complex observable phenomena.

A review by Speybroeck and colleagues, covering research published before January 2013, explored how simulation models had been used in the field of socioeconomic inequalities in health specifically [14]. They found only four ABM studies, which focused on understanding differences in health behaviour or infectious disease transmission between socioeconomic groups. Speybroeck and colleagues concluded that ABM is the most appropriate computational modelling method to examine health inequalities as they can incorporate all the characteristics of a complex system such as the heterogeneity, interactions, feedback, and emergence [14]. However, while the four identified models contained many of the expected features of ABM (e.g., multi-level, dynamic, and stochastic), the Speybroeck review concluded that to better understand the complex mechanisms underlying health inequalities, more ABM that features feedback loops, temporal changes, and agent–agent and agent–environment interactions are required.

Since the Speybroeck review, there has been a methodological shift towards using complex system methods in public health and public policy, much supported by large investments in data accessibility and computing power. In the UK, this is also reflected in the Medical Research Council’s updated guidance for the development and evaluation of complex interventions [15] and the Her Majesty Treasury’s Magenta Book Annex “Handling Complexity in Policy Evaluation”, both published in 2021 [16]. This methodological turn has resulted in a significant increase in computational modelling papers in the public health literature in recent years; therefore, it is now timely to update and deepen the previous review. Here, we focus on the contribution of ABM to understand the socioeconomic inequalities in health specifically, by reviewing the application area (e.g., the inequality mechanisms studied, the choice of the health outcome(s), and the measure of socioeconomic position), and the details of the ABM approach (e.g., the represented complexity features and whether models have been validated). The aim of this review was to synthesise the growing evidence based on the use of ABM in the field of health inequalities research.

## 2. Materials and Methods

We followed the guidelines of the Preferred Reporting Items for Systematic Reviews and Meta-Analyses (PRISMA) [17]. The protocol for this review was developed and registered on the International Prospective Register of Systematic Reviews (protocol registration PROSPERO 2022 CRD42022301797). PubMed, Scopus, and Web of Science were searched from 1 January 2013 to 15 November 2022. The Scopus search was limited by subject area to Medicine; Social Sciences; Computer Science; Multidisciplinary; Mathematics; Nursing; Economics, Econometrics and Finance; Neuroscience; Health Professions; Psychology; Decision Sciences; and Engineering. For Web of Science, searches were made of the editions of Science Citation Index Expanded and Social Sciences Citation Index. For both Web of Science and PubMed, only the titles and abstracts were searched. An extensive list of search terms was used (see Appendix A) to capture the themes of simulation modelling, socioeconomic inequality, and health. The search strategy was validated against that used in the Speybroeck review [14], confirming that all ABM studies included in that review also appeared using our search strategy.

### 2.1. Eligibility Criteria

Table 1 lists the inclusion criteria for this review; this criterion includes the population, exposures, comparisons, outcomes, and study designs (PECOS) required for a study to be eligible for inclusion. Studies were included if they: (i) were full papers published in English, and (ii) the paper described an ABM study with the purpose to understand the emergence and/or persistence of health inequalities in relation to either non-communicable disease or the differential response of different socioeconomic groups to health-related interventions. Papers were only included if they simulated human individuals or groups and investigated within-country socioeconomic inequalities (using measures such as the socioeconomic position, income, and education) in health, restricted to the differences in the health status, health behaviour, or access to healthcare. Papers in which healthy food access was modelled as a proxy for the consumption of healthy food were also included. Studies that developed ABM in combination with system dynamics or population-based models were included. There were no geographical restrictions.

Papers that modelled communicable diseases or water or food access/security as the health outcomes were outside the scope of this review and were therefore excluded. The studies published before 2013 were also excluded as these studies were covered in the Speybroeck review [14].

### 2.2. Screening

Searching returned a total of 2533 records. All the records were downloaded to EndNote X9 and imported to the EPPI-Reviewer. The total records were reduced to 1436 following the removal of duplicates. An initial screening was carried out by one reviewer (RW). Following title screening, 477 records were identified for abstract screening. A second reviewer (JB) independently double-screened a randomly selected subset of abstracts (20%). After title and abstract screening, 51 records were selected for full-text screening and 18 of these met the eligibility criteria for data synthesis (Figure 1). The second reviewer (JB) also independently screened all the selected full-text studies to validate that the included papers met all the eligibility criteria. Any disagreements were recorded and discussed to ensure consistency. Two further reviewers (CE and AH) assisted with the screening for papers queried on methodological grounds (*n* = 29), in instances where it was uncertain whether a simulation model met the inclusion criteria. Manual reference searching identified two additional papers which met the inclusion criteria, giving a final sample of 20 included studies.

### 2.3. Data Extraction

Data from the papers were extracted by one reviewer (RW). A second reviewer (JB) assessed the accuracy of the data extraction for all the included studies. In the case of a disagreement, both reviewers referred to the paper in question, and a consensus was reached. A data extraction matrix was developed which included the basic characteristics of the studies (the year, location, and study’s aims), variables modelled (socioeconomic measure and health outcome), model characteristics (multi-level, dynamic, feedback loop, stochastic, spatial, heterogenous, agent–agent interaction, and adaptation to environment), if and how the model was validated, the model’s function (framework development and/or to test an intervention/scenario), and the relevant findings. The model’s characteristics were not always explicit but could be derived from the methods section. The relevant findings were defined as those related to health or intervention outcomes stratified by a measure of the socioeconomic position.

### 2.4. Quality Assessment

Given the lack of an appropriate quality assessment or a risk of bias assessment tool to assess ABM, a quality assessment was not conducted, but we recorded the compliance with the reporting guidelines of the ODD (the overview, design concepts, and details) [18].

### 2.5. Analysis

Descriptive summary statistics were used to describe the search results and study’s characteristics. We describe the specific modelling details of the included studies using a narrative synthesis in which we group models based on the health outcome.

## 3. Results

### 3.1. Descriptive Analysis

The study characteristics for the 20 included papers are displayed in Table 2. The most common geographical settings for the models were the USA (*n* = 7) and the UK (*n* = 4). The other models were set in the Netherlands, Mexico, India, South Korea, Canada, and Japan. Only two models were abstract and did not have a geographical setting. Most of the included models were set at the city level (*n* = 10), other settings included the national (*n* = 5), state (*n* = 2), and district level (*n* = 1).

Most of the included papers described the ABM of the socioeconomic differences in health behaviour (*n* = 14). Three papers focused on explaining the socioeconomic differences in the physical health outcomes and three papers modelled a mental health outcome. The measures of the socioeconomic position covered the income (*n* = 14), educational attainment (*n* = 4), social grade (*n* = 2), and wealth (*n* = 1).

All of the included models were multi-level (they represented both individuals and structural entities), dynamic (captured changes over time), stochastic (based on probabilities), and had heterogeneous agents. Most models represented both the individuals and the environment with environmental features (e.g., shops, green spaces, and public transport). Often, in the models, agents could age, die, and change their behaviour over the course of the simulation. Only three papers used the ODD reporting guidelines when writing descriptions of their ABM [18].

### 3.2. Health Behaviours

Most models with a focus on the health behaviour modelled dietary behaviours (*n* = 7). Four of the models were concerned with physical activity and the access to green space, and three modelled substance use, specifically the consumption or purchase of alcohol and tobacco as a proxy for consumption.

#### 3.2.1. Dietary Behaviour

Papers that used ABM to model the socioeconomic differences in dietary behaviours tested the impact of interventions on the consumption of sugar-sweetened beverages [19], the purchase of ultra-processed food [20], the consumption of fruits and vegetables [21,22], and the access to healthy food outlets [23]. The interventions were educational campaigns (e.g., nutrition warnings and school-based programmes), advertising campaigns, changes to tax, increasing access to vegetables, and reducing the cost of vegetables. However, two papers focused on the impact of residential segregation on the access to healthy food outlets as an explanation for the socioeconomic differences in dietary behaviours [24,25]. All models used the level of income of the individual or household, educational achievement, or both as the measure of the socioeconomic position.

The only paper that did not include a spatial component to the model was set at a national level, and explored the impact of tax, nutrition warnings, and advertising on the purchase of ultra-processed food in Mexico [20]. The other six models used artificial grid space [24], a 1-dimensional linear township [25], a raster map to represent the spatial distribution of income [21], or actual geographic space, including GIS modelling of real-life cities [19,22,23]. Six of the models included agent–environment interactions which often captured how individual agents engage with food outlets [21,22,23,24,25]. Only two of the included papers modelled agent–agent interactions through dietary social norms operationalised via a social network which influenced the taste preferences and health beliefs [22], and the purchasing of ultra-processed foods [20]. Five of the models featured feedback loops, these included the update of social norms based on behaviour over the course of the simulation [20,22], and the food outlet responses to the agent’s behaviour by closing and opening outlets [23], changing the type of food available for sale [24], or an increasing appetite and overeating following the consumption of foods high in fat, sugar, and salt [21]. Only two of the papers had attempted validation by comparing the simulated outcomes to the ‘observed’ outcomes in real world data [19,22].

#### 3.2.2. Physical Activity and Use of Urban Green Space

All the models that investigated the socioeconomic differences in physical activity simulated intervention scenarios. These scenarios included additional physical education in schools, the promotion of active travel, educational campaigns, increasing the availability and affordability of sports activities, improving neighbourhood safety, and increasing the expense associated with driving [26,27,28]. All the models focusing on physical activity used the level of income of the individual or household as the measure of the socioeconomic status and explored a range of physical activity-related outcomes including the minutes of physical activity per day [26], sports participation [27], and walking [28]. Models concerning physical activity involved a spatial component operationalised as either a representation of the actual geographical space [26,27] or an artificial grid [28]. All the models simulated both agent–agent interactions (e.g., social interactions modelled via a social network that impacts behaviour) and agent–environment interactions (e.g., playing outdoors or engaging with sports facilities in the environment). Two models contained feedback loops, including the updating of social norms regarding exercise and travel preferences [27,28] and environmental feedback, including the safety and traffic levels of travel routes on the attitudes towards transport methods [28]. Two models were validated by comparing the simulated outcomes to the outcomes observed in the pre-existing data [26,28].

One paper modelled intra- and inter-city inequalities in visiting urban green spaces, specifically testing the mechanism that the decision to visit these spaces is influenced by an individual’s assessment of who had previously visited the space [29]. Given conflicting evidence, the model explored two possibilities: (1) that agents visit spaces that people like themselves to visit (homophilic preference) and (2) that individuals with a lower SES (socioeconomic status) prefer to spend time in areas which those of a high SES visit (heterophilic preference). This model used the occupational grade to classify the agents into either a high or low SES. The model spatially represented the cities of Edinburgh, Glasgow, Aberdeen, and Dundee, and simulated both agent–environment interactions in the form of visiting urban green spaces and agent–agent interactions via agents assessing the similarity of other agents visiting the green space. The feedback loop in this model was the update of who visited green spaces, which was a function of the update to whether ‘in-’ or ‘out-’ group members were present in those spaces. Given a lack of observed data, the model was validated using a pattern matching approach; the model could reproduce the observed patterns of urban green space visitation in a spatial microsimulation of Glasgow.

#### 3.2.3. Substance Use

Two of the models that focused on the socioeconomic differences in substance use tested the impact of interventions, including alcohol taxation [30] and the restriction of menthol cigarette sales and tobacco retailer density [31]. One paper simulated several counterfactual scenarios which varied the degree of socioeconomic disparity and gender-related susceptibility to social influence in the context of smoking [32]. All the models used the income level of the individual or household as the measure of the socioeconomic position and investigated substance use in the form of smoking prevalence [32], tobacco purchasing [31], and the average number of alcoholic drinks per day [30].

Two models simulated agent–agent interactions including the influence of gender and socioeconomic social norms on an individual’s own smoking [32], and social network influences on drinking behaviour [30]. Two models were spatial; one represented the city of New York [30] and the other an abstract town called ‘Tobacco Town’ [31]. Two models simulated agent–environment interactions, such as travelling to and from locations and engaging with tobacco and alcohol retail outlets [30,31]. One paper not only focused on the consumption of alcohol but also examined the interaction between neighbourhood characteristics, social networks, sociodemographic characteristics, drinking, and violence [30]. Two models featured feedback loops in the form of updates to norms based on drinking and smoking behaviour [30,32]. One model validated the simulated outcomes by comparing these to the outcomes observed in real-world data on the prevalence of smoking in Japan [32].

### 3.3. Physical Health

Of the three models that focused on physical health outcomes, one examined the incidence of severe neonatal morbidity and deaths per 1000 live births averted [33], one looked at the health status and care need [34], and the other investigated the impact of an exposure to air pollution on the health status [35]. Two of the papers modelled the effect of potential interventions on the physical health outcomes [33,34]. In one, the intervention was the alteration of the eligibility criteria for government-funded social care, in the other increasing the responsibilities and coverage of community health workers. All the models used a different measure of the socioeconomic position including wealth quintile [33], approximated social grade [34], and educational attainment [35].

All three models included the individual and household levels and two included additional levels such as kinship networks and the regional level. One study represented space using a grid based on the geography of the modelled country [34] and two represented the actual geographic space [33,35]. An interaction with the environment was in the form of migration, seeking treatment at facilities, and the exposure to pollution. Only two models included a feedback loop, including feedback between the parental income level and childhood educational attainment and feedback between the level of disease and the probability of developing a further disease [34,35]. Only one model involved agents interacting with each other, in the form of a kinship network, which consisted of familial relationships [34]. None of these models validated their results using real world data. One of the models was used to create a complex theoretical framework to represent the social care system. The geographical and population data inputted into this framework could then be adjusted to model and understand the drivers of the unmet social care need in different countries [34].

### 3.4. Mental Health

Two of the three papers focusing on a mental health outcome examined the impact of transport on depression among older adults. The first examined the impact of multiple transport interventions [36], and the second examined that of a free bus policy on public transit use and depression [37]. An individual’s income was used as a measure of the socioeconomic status in both papers.

One model carried out three experiments: increasing the walkability and safety of neighbourhoods to promote walking; decreasing bus fares and bus waiting times; and adding bus lines and stations [36]. While the second model focusing on transport carried out four experiments: altering mean attitudes towards the bus; bus waiting times; the cost of parking; and fuel prices; each experiment was also carried out with and without the inclusion of the free bus policy [37]. Both models captured the individual and neighbourhood level. A feedback loop resulted in improved attitudes towards a certain mode of travel following the positive experience of that mode. The spatial element was applied to income segregation patterns. In one model, the agents interacted with each other in the form of social networks influencing the travel behaviour [37]. In both models, agents interacted with the environment by using transport. Both models were validated against empirical data on the prevalence of depression in the United States by gender, age, and income level.

The third paper examined the impact of reducing income inequality on depression among expectant mothers [38]. Four interventions to increase income were tested: two child benefit programs (ACB and CCB), universal basic income (UBI), and increasing minimum wage. This model focused on individuals, and while it captured the neighbourhood characteristics for each individual (e.g., a sense of safety and the prenatal services available), the environment was not spatially represented in the model. Agents could decide to make or break social connections with other agents, and whether to break ties with other agents with depression. This model was not validated.

### 3.5. What Can ABM Tell Us about Socioeconomic Inequalities in Health?

Studies investigating the explanations for the socioeconomic differences in health found that those of a higher socioeconomic position were more likely to be exposed to healthier environments and therefore engage in healthier behaviours and have better health outcomes. For example, one model found that a greater income segregation in communities led to a decreased access to healthy food for lower income households [25]. Another study which modelled agents’ movements from work to home found that regardless of the level of air pollution, those with a lower level of education consistently had the highest risk of developing an illness [35].

Models which tested the impact of interventions on the socioeconomic inequalities in health found that some interventions increased inequalities. For example, those of a high socioeconomic position improved their health behaviour more in response to educational campaigns concerning nutrition [22,23]. It was argued that nutritional education campaigns may be ineffective for those of a lower socioeconomic position due to a sensitivity to food prices and a lack of access to healthy alternatives [22]. Similarly, it was found that the promotion of active travel had greater benefits for those of a high socioeconomic position, as they are more likely to travel by car and travel by car more often to extra-curricular activities prior to the intervention [26].

However, there were some modelled interventions that decreased the socioeconomic inequalities in health. For example, one model tested the impact of a sugar-sweetened beverage tax and found that at 25% tax, the reduction in the consumption of sugar-sweetened beverages was greater in those from low-income populations [19]. This finding was largely the result of increases in price which made sugar-sweetened beverages less affordable to low-income households. Another study, which modelled the expanded responsibilities and increased coverage for accredited social health activists who perform postnatal check-ups, found that these interventions resulted in greater decreases in the neonatal morbidity and mortality among those of a low socioeconomic position [33]. Yang and colleagues also showed that, in older adults when attitudes towards bus use improved and the waiting time decreased, decreases in depression were estimated to be greater among low-income groups [37]. This larger increase was because those of a low income are less likely to own cars and are therefore more susceptible to an intervention to increase the uptake of public transport, which increases the number of non-work trips they take, which was beneficial to their mental health.

## 4. Discussion

This review included 20 papers that described the ABM of the socioeconomic inequalities in health that have been published since January 2013, the end point of the Speybroeck review which found only four ABM studies on the topic [14]. Using ABM in the context of socioeconomic health inequalities was most common in the USA and UK (*n* = 11). The included studies illustrated that ABM is a useful tool to understand complex problems and has been used flexibly to represent dynamic, multi-level processes, often in physical space, and to capture the interactions between individuals and interactions with their environment. These models can tell us about the causes of health inequalities, potential interventions to reduce health inequalities, and which interventions may inadvertently increase health inequalities.

Typically, ABM has been used to explore socioeconomic differences in health behaviours (*n* = 14) including diet, physical activity, access to green space, and substance use, but few have approached the socioeconomic differences in physical and mental health outcomes. Additionally, only one paper modelled access to healthcare as a potential explanation for socioeconomic inequalities in health [33]. To an extent this is unsurprising, given a historic focus on health behaviours in public health [39] coupled with the fact that ABM as a method captures how behaviours at the micro-level give rise to emergent phenomena at the macro-level [40].

Most ABM were used to test a range of interventions (*n* = 14), from educational campaigns to taxation, and were underutilised for other purposes, such as testing the explanatory value of the theory or mechanisms to explain the generation or persistence of socioeconomic inequalities in health. This is consistent with the Speybroeck review which found that all ABM studies were used to test an intervention or scenario [14] and highlights that a valuable feature of ABM is the ability to experiment and test a range of interventions in silico [40].

Less than half of the included studies (*n* = 9) attempted to validate their models and, to varying degrees, some using observational data or pattern matching methods. However, none of the included studies used structural validation techniques which would ensure that it is the intended “structure of the model that drives its behaviour” [41]. This finding is consistent with the Speybroeck review which found that only one ABM had been validated using observational data [14]. Additionally, only three of the included papers explicitly used and referred to the ODD protocol, the guidelines with the purpose of ensuring that ABM is described fully to facilitate its replication [18].

It is clear from the findings of this review that most existing ABM studies investigating the socioeconomic inequalities in health have focused on health behaviour. This individualistic focus on health inequalities in ABM efforts on this topic so far is not reflective of ABMs in the field of social science more generally. ABM has been used to understand broader social phenomena such as racial segregation and the generation of wealth inequality at the societal level [11,12,13]. While these patterns are generated by individual-level behaviours, these models do not seek to explain these behaviours. Reducing health inequality to understanding the differences in health behaviour is problematic given that research has shown that for the same level of any given behaviour, the health outcomes remain worse for the most socioeconomically deprived [42].

### 4.1. Limitations

Currently there is no available tool to assess the quality of ABM studies, and therefore we could not ensure that the models included in this review were of a high quality. There are a variety of quality assessment tools available to assess other study types, for example, the appraisal tool for cross-sectional studies (AXIS) which can be used to assess a study’s design, reporting quality, and risk of bias [43]. Given the increase in ABM studies in public health, it is critical to consider how we will assess the quality of these studies going forward.

While the Speybroeck review considered a breadth of simulation models [14], we chose to focus on ABM only, given the particular promise of ABM applied to health inequality research and the rapid increase in the use of simulation modelling techniques since 2013 [10]. The application of alternative simulation modelling techniques (e.g., microsimulation and system dynamics) to the socioeconomic inequalities in health in recent years awaits a further examination.

### 4.2. Future Research

Efforts thus far to use ABM to understand socioeconomic inequalities in health have focused on the contribution of health behaviour. However, this focus on health behaviour is at odds with calls from researchers to “move beyond bad behaviours” [44] and the position of influential public health organisations. For example, the WHO concluded that it is the underlying social and economic factors that determine health and health inequalities as opposed to health behaviours [45]. We are increasingly aware that health inequalities are not only the result of differences in health behaviour, yet little has been done using ABM to attempt to understand the complex relationships between the social and economic environment people live in and the influence on their health via pathways other than health behaviour.

There are explanations for socioeconomic inequalities in health that shift the focus from individual-level behaviours to the social determinants which themselves determine health and to an extent behaviour (e.g., the social determinants of health) [45]. Existing ABMs have started to look at the social drivers of health behaviours (e.g., the role of social network and social norms) [20,22], however they avoid alternative pathways through which social and economic factors directly or indirectly impact health. It has been argued that ABM could be used to investigate the mechanisms specified in social and economic explanations for health inequality [46]. An existing hypothetical example of how this may be done is the operationalisation of psychosocial theory [46]. Instead of a focus on health behaviours, operationalising psychosocial theory would involve simulating support seeking and giving among friendship networks which mediates health outcomes via stress pathways. Future research should consider how we can use ABM to simulate alternative mechanisms which could explain the socioeconomic inequalities in health that are not exclusively focused on health behaviour.

## 5. Conclusions

In recent years, ABM has increasingly been used to explain socioeconomic inequalities in health. ABM allows us to develop a deeper understanding of the complex consequences of individual heterogeneity, spatial settings, feedback, and adaptation resulting from agent interactions with each other and their environment. However, to date, much of the focus has been on understanding the role of health behaviours. The features of ABM provide the opportunity to investigate alternative, more complex explanations for socioeconomic health inequalities. Therefore, an important next step in public health is to attempt to operationalise explanations for the causes and consequences of health inequalities beyond representations of health behaviour.

## Figures and Tables

**Figure 1 ijerph-19-16807-f001:**
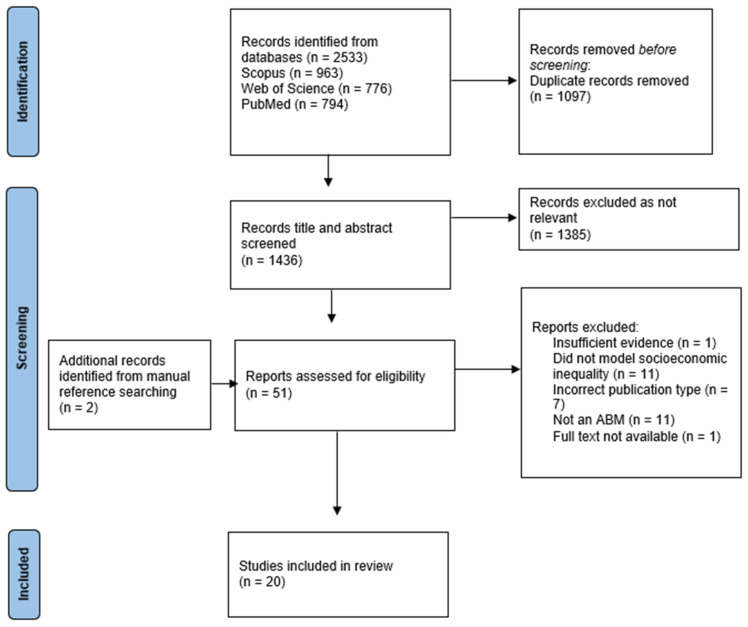
PRISMA Flow Diagram.

**Table 1 ijerph-19-16807-t001:** Population, exposures, comparisons, outcomes, and study design criteria for study inclusion.

Criteria	Definition
Population	Human individuals or groups
Exposures	Socioeconomic position (any measure including income, educational status, occupation, and employment status)
Comparisons	Compares individuals to other individuals with a different level of socioeconomic position (e.g., high vs. low)
Outcomes	Health (any measure that captures health status, health behaviour or access to healthcare)
Study design	Only ABM studies were included

**Table 2 ijerph-19-16807-t002:** Characteristics of included papers.

Author, Year	Country of Authors	Population	Aim of Study	SE Factor Modelled	Health Outcome Modelled	Characteristics of the Model								Validation and Utilisation of Model		
						ML	D	St	FL	Sp	HtI	AI	EI	V	F	I
Almagor et al., 2021	UK	Glasgow	To explore the potential impact of interventions on physical activity of children in an urban environment.	SEP divided into 4 levels representing a gradient of household income: AB-high, C1, C2, DE-low.	Minutes of moderate-to-vigorous physical activity/day.	✕	✕	✕		✕	✕	✕	✕	✕		✕
Auchincloss and Garcia, 2015	USA, Brazil	Abstract Space	Introduce guide for agent-based modelling and explore impact of urban segregation on inequalities in diet.	Urban segregation by household income—location and income of households (divided into low or high-income).	Average proportion of times the household shopped at a healthy food store (depends on household income, proximity to stores, and food preferences).	✕	✕	✕	✕	✕	✕		✕			
Benny et al., 2022	Canada	Calgary	To simulate the effects of government transfers and increases to minimum wage on depression in mothers.	Individual Income categorised into CAD 39,999 or less, CAD 40,000 to 79,999, and CAD 80,000 or more. Education categorised into high school or less, some or completed university/college, and some or completed graduate school.	Depression measured using the Edinburgh Postnatal Depression Scale (EPDS).	✕	✕	✕			✕	✕				✕
Blok et al., 2015	The Netherlands	Eindhoven	To explore the impact of 3 interventions (eliminating residential income segregation, reducing prices of healthy food, health education) aimed at reducing food consumption inequalities between low and high-income households.	Household Income divided into high (>USD 31,777/year) and low (<USD 31,777/year).	Average proportion of times a household visited a healthy food outlet.	✕	✕	✕	✕	✕	✕		✕			✕
Blok et al., 2018	The Netherlands	Eindhoven	Explore impact of 5 interventions (health education, lowering prices of sports facilities, increasing availability of sports facilities, improving neighbourhood safety, combining all these interventions) on reducing income inequalities in sports.	Individual Income divided into low, middle, and high.	% of individuals participating in sport annually.	✕	✕	✕	✕	✕	✕	✕	✕			✕
Chao et al., 2015	Japan	Japan	Explore how socioeconomic disparity between and within gender groups affects changes in smoking prevalence.	Socioeconomic Status divided into 1–9 according to distribution of income.	% of each gender group who were smoking.	✕	✕	✕	✕		✕	✕		✕		
Combs et al., 2020	USA	Tobacco Town, Minnesota	Project the impact of menthol cigarette sales restrictions and retailer density reduction policies on tobacco sales for low income, African American and LGBTQ+ populations	Individual Income, divided into two groups: low-income (<USD 42,500) and high-income (>USD 42,500).	Costs to consumers per pack of cigarettes as proxy for tobacco consumption.	✕	✕	✕		✕	✕		✕			✕
Gostoli and Silverman, 2019	UK	UK	Provide theoretical framework to understand drivers of unmet social care need and test policies.	Approximated Social Grade, a socioeconomic classification produced by the Office for National Statistics (six categories A, B, C1, C2, D, and E).	Health status and care need (weekly hours of care required); death (affected by agents’ level of unmet care need).	✕	✕	✕	✕	✕	✕	✕	✕		✕	✕
Gouri Suresh and Schauder, 2020	USA	USA	Explore how income segregation impacts food access for poor, when preferences and knowledge of healthy foods are equal among different income groups.	Household Income randomly generated on the basis of 2016 income distribution reported by US census bureau.	Distance to nearest grocery store and whether healthy food was reliably available at nearest grocery store.	✕	✕	✕		✕	✕		✕			
Keyes et al., 2019	USA	New York City	Estimate the impact of alcohol taxation on drinking, violence and homicide.	Household Income—stratified into 5 quintiles.	Average number of alcoholic drinks per day.	✕	✕	✕	✕	✕	✕	✕	✕			✕
Langellier et al., 2017	USA	Philadelphia	Evaluate impact of beverage tax and pre-kindergarten programme on children’s SSB consumption.	Household Income—categorised as low-income (≤100% of Federal Poverty Level) and modest-income (≤300% of FPL) households.	Sugar Sweetened Beverage consumption in drinks/week.	✕	✕	✕		✕	✕		✕	✕		✕
Langellier et al., 2021	USA, Australia, Brazil	Mexico	Develop a simulation framework to assess how tax, nutrition warning and advertising impact ultra-processed food purchasing.	Individual Income, divided into low-income (<1890 pesos/week) and high-income (>1890 pesos/week). EA, divided into low- (less than high school education) and high-education (at least high school).	Ultra-processed food purchased, measured in kcal (energy intake) purchased per week.	✕	✕	✕	✕		✕	✕			✕	✕
Li et al., 2016	USA	New York City	Simulate how mass media and nutrition education change fruit and vegetable consumption in NYC.	Educational Attainment, categorised by less than high school, high school, some college and college and above.	Proportion of the population in a given neighbourhood who consume on average >2 servings of fruit and vegetables per day.	✕	✕	✕	✕	✕	✕	✕	✕	✕		✕
Nandi et al., 2016	USA, India, UK	India	Estimate reduction in disease burden by scaling up home-based newborn care in rural India.	Wealth quintile.	Incidence cases of severe neonatal morbidity averted and deaths per 1000 live births averted.	✕	✕	✕		✕	✕		✕			✕
Picascia and Mitchell, 2022	UK	Edinburgh, Dundee, Glasgow, Aberdeen	Investigate intra- and inter-city inequalities in Urban Green Spaces visiting by SES.	SES divided into 4 categories based on occupational grade: AB-high, C1, C2, DE-low.	Median number of visits to urban green space/year.	✕	✕	✕	✕	✕	✕	✕	✕	✕		
Salvo et al., 2022	USA	Austin, Texas	To simulate the food environment and test the impact of different food access policies on vegetable consumption.	Annual Household Income categorised into: Under USD 25,000, USD 25,001–USD 45,000, USD 45,001–USD 65,000, and >USD 65,000, and educational attainment in four categories: <High school, High school or GED, Some college, and Full college or more.	Fruit and vegetable intake.	✕	✕	✕	✕	✕	✕		✕			✕
Shin and Bithell, 2019	UK	Gangnam and Gwanak districts, Seoul	Understand cumulative effects of PM10 exposure on population vulnerability by education level and age.	Educational Attainment in 8 categories: primary-school dropout, primary-school graduate, middle-school dropout, middle-school graduate, high-school dropout, high-school graduate, college or university student, over a bachelor’s degree.	Health status: starts with 300 and drops when exposed to pollution; categorised into <100, 100–150, or 150–200.	✕	✕	✕	✕	✕	✕		✕	✕		
Yang et al., 2015	USA	US city	Explore how travel costs and educational interventions can alter income differentials in walking.	Household Income segregation—income divided into quintiles (1 to 5).	Proportion of trips to destinations made by walking.	✕	✕	✕	✕	✕	✕	✕	✕	✕		✕
Yang et al., 2019	USA	Abstract space	Investigate how transport interventions may affect depression in older adults.	Individual Income—divided into quintiles (1 to 5).	Depression status yes/no, where having depression is a score of >/=4 on CESD Scale-8.	✕	✕	✕	✕	✕	✕	✕	✕	✕		✕
Yang et al., 2020	USA, UK, The Netherlands	English city	Examine the impact of a free bus policy on public transit use and depression among older adults.	Individual Income—divided into quintiles (1 to 5).	Prevalence/ percentage of agents with depression.	✕	✕	✕	✕	✕	✕	✕	✕	✕		✕

ML—multi-level. D—dynamic. St—stochastic. FL—feedback loop. Sp—spatial. HtI—heterogeneous individuals. AI—agent–agent interactions. EI—agent–environment interactions. V—validation. F—framework. I—test an intervention.

## Data Availability

Not applicable.

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
