# Peer review of "Agent-Based Modelling of Health Inequalities following the Complexity Turn in Public Health: A Systematic Review"

_ijerph, 2022, doi:10.3390/ijerph192416807_

Round 1

Reviewer 1 Report

This article looks at a review of health research conducted in recent years using Agent Based Modeling.  I think this is an appropriate topic to use as a call more additional research on the topic.  However, I do not think the authors did a sufficient review of the research.  I understand they limited their parameters for recent research, but I think a more exhaustive research, adding some seminal works on Agent Based modeling would help demonstrate the strengths and weaknesses of this approach.  This would include work from outside of public health and the medical fields.  Agent Based Modeling has a robust history in the social sciences, with much research looking at the strengths and weaknesses of the approach.  I think it would benefit other scholars reading your work to address the assumptions, challenges and benefits of Agent Based Modelling.  

Thank for you this article.  I enjoyed reading it.  Best of luck in future research.  

Author Response

Many thanks for your review, please see attached our response.

Reviewer 2 Report

The paper is well written.  In the abstract section sentence: "Systematic review (Protocol registration PROSPERO 2022 CRD42022301797)" must be changed and incorporated in the text because it is not sentence. Also in the method section regarding eligibility criteria  it would be useful to define including and excluding criteria for the studies, not to be mixed troughout the text.

Author Response

(The authors gave the same response as above.)

Reviewer 3 Report

Thank you for the opportunity of reading this paper. 

The abstract comprises the main idea of the paper, the introduction is sufficiently detailed, still, I would recommend an extension in terms of theoretical background.

The Discussion and the Conclusions sections must be extended, they are currently insufficiently developed.

I recommend an update of the bibliographic sources, as half of the references ar eolder than 2018.

Author Response

(The authors gave the same response as above.)

Reviewer 4 Report

This paper has made a systematic review of the papers about agent-based modelling of health Inequalities. The research perspective is interesting with scientific and systematic research design and process. The findings has contributed to more updated understanding about the socioeconomic inequalities in health focusing on health behaviour, which help to extend the discussion of the Speybroeck review since 2013. Meanwhile, the work has been well organized and the format meets well the requirement of the journal ijerph.

Thus, I recommend to accept this paper in present form.

Author Response

We would like to thank the reviewer for their kind comments on our paper.

Round 2

Reviewer 1 Report

Thank you for the second draft of the paper.  I also thank you for addressing my suggestions.  I think the small edition will significantly improve contextualizing the research and helping readers understand the value of agent based modeling.  I have no real concerns with the paper as it is currently written.  Best of luck with your future research.  

Author Response

Thank you for your review, we are glad we have been able to address all concerns.